# Short and Long-Term Wellbeing of Children following SARS-CoV-2 Infection: A Systematic Review

**DOI:** 10.3390/ijerph192114392

**Published:** 2022-11-03

**Authors:** Juan Victor Ariel Franco, Luis Ignacio Garegnani, Gisela Viviana Oltra, Maria-Inti Metzendorf, Leonel Fabrizio Trivisonno, Nadia Sgarbossa, Denise Ducks, Katharina Heldt, Rebekka Mumm, Benjamin Barnes, Christa Scheidt-Nave

**Affiliations:** 1Institute of General Practice, Medical Faculty, Heinrich-Heine-University Düsseldorf, 40225 Düsseldorf, Germany; 2Research Department, Instituto Universitario Hospital Italiano de Buenos Aires, Buenos Aires 4234, Argentina; 3Department of Health Science, Universidad Nacional de La Matanza, Buenos Aires 1754, Argentina; 4Department of Epidemiology and Health Monitoring, Robert Koch-Institute, 13353 Berlin, Germany

**Keywords:** COVID-19, post-COVID-conditions, children, public health

## Abstract

Post-COVID conditions in children and adolescents were mostly investigated as the incidence of individual or clusters of symptoms. We aimed to describe the findings of studies assessing key outcomes related to global wellbeing and recovery in children and adolescents from a public health perspective. We searched the Cochrane COVID-19 Study Register and WHO COVID-19 Global literature on coronavirus disease database on 5 November 2021 and tracked ongoing studies published after this date. We included observational studies on children and adolescents with a follow-up greater than 12 weeks and focused on the outcomes of quality of life, recovery/duration of symptoms, school attendance and resource use/rehabilitation. We assessed their methodological quality, and we prepared a narrative synthesis of the results. We included 21 longitudinal and 4 cross-sectional studies (6 with a control group) with over 68 thousand unvaccinated children and adolescents with mostly asymptomatic or mild disease. Study limitations included convenience sampling, a poor description of their study population and heterogeneous definitions of outcomes. Quality of life was not largely affected in adolescents following COVID-19, but there might be greater impairment in young children and in those with more severe forms of the disease (4 studies). There might also be an impairment in daily activities and increased school absenteeism following COVID-19, but the findings were heterogeneous (5 studies). A total of 22 studies provided highly variable estimates based on heterogeneous definitions of overall persistence of symptoms (recovery), ranging from 0 to 67% at 8–12 weeks and 8 to 51% at 6–12 months. We found limited data on resource use and the need for rehabilitation. One controlled study indicated that the quality of life of infected children and adolescents might not substantially differ from controls. All controlled studies found a higher burden of persistent symptoms in COVID-19 cases compared with test-negative controls or cases of influenza. There is limited evidence on the short and long-term well-being of children following SARS-CoV-2 infection. High-quality longitudinal studies with control groups are needed to describe the outcomes in this population, especially in vaccinated children and those affected by new variants of the virus.

## 1. Introduction

The spread of the novel coronavirus designated as Severe Acute Respiratory Syndrome Coronavirus 2 (SARS-CoV-2) triggered a pneumonia and systemic disease outbreak called Coronavirus disease 2019 (COVID-19) which spread throughout the world in early 2020 [1]. While most infected people have mild disease with nonspecific symptoms, approximately 5% of patients with COVID-19 experience severe symptoms and become critically ill [2]. Age is an important risk factor for severe disease, and elderly adults are particularly vulnerable. Conversely, morbidity and mortality are lower in children and adolescents. Out of the 3.4 million deaths due to COVID-19, approximately 13,000 occurred in children and adolescents below the age of 20 [3].

Beyond the acute phase, Long COVID is usually used as a term to describe the persistence or recurrence of health symptoms beyond the acute phase of infection, considered to extend to four weeks [4,5]. The World Health Organisation (WHO) developed a clinical case definition of post-COVID-19 conditions, including persistent, relapsing or new symptoms beyond 12 weeks after infection [6]. A more recent research definition was derived for children and young people, including similar diagnostic criteria and timeframe and focusing on limitations of routine functioning [7]. While it is clear that persistent symptoms of COVID-19 can be present in children and adolescents, it is unclear what proportion of those infected suffer from this long-term condition, as well as what its course and prognosis are [8,9,10,11]. There is a high variety of duration of symptoms, ranging from 14 days [9] up to more than one year [11,12]. Reported symptoms are also highly heterogeneous, covering a vast extension from loss of smell or taste [13] to resting dyspnea, palpitations or tachycardia [14] as well as musculoskeletal, mental health and neurocognitive symptoms.

Despite the favourable outcomes for children and adolescents during the acute stage of the disease, long-term consequences of the infection have been reported [15]. Lopez-Leon et al. conducted a systematic review to estimate the prevalence of Long COVID in children and adolescents, including 21 studies with a prevalence of Long COVID of 25%. Mood changes, fatigue and sleep disorders were the most common manifestations [16]. Behnood et al. also conducted a systematic review which reported a mean duration of symptoms of 125 days and a Long COVID prevalence ranging between 15% and almost 50%. Long COVID patients were more likely to experience cognitive difficulties, headaches, loss of smell, sore throat or sore eyes [17].

Our team has developed an evidence map of observational studies analysing long-term symptoms and sequelae following SARS-CoV-2 infection, which is available on the website of the Robert Koch Institute (www.rki.de/post-covid-evimaps, accessed on 25 September 2022). Up to November 2021, we found few studies that assessed the long-term course and prognosis of symptoms related to Long COVID in children and adolescents. Most studies focused on the prevalence of individual symptoms (e.g., anosmia, fatigue, etc.). In this systematic review, we aim to describe the findings of studies assessing key outcomes related to wellbeing and recovery in children and adolescents using the evidence derived from our evidence map [18].

## 2. Materials and Methods

This systematic review is based on a previously developed evidence map aimed to collect the available evidence on persistent symptoms and sequelae following SARS-CoV-2 infection in children and adults [18]. We followed the Joanna Briggs Institute (JBI) guideline for systematic reviews of prevalence studies [19], including the guideline of our predefined protocol registered in OSF (osf.io/tkqes) [20]. We reported the findings following the PRISMA guideline for systematic reviews [21].

### 2.1. Inclusion Criteria

Type of studies: We included observational studies (longitudinal and cross-sectional), including those embedded in randomised controlled trials with 12 or more weeks of follow-up time. We excluded case reports, case series and those only focusing on people with sequelae or persistent symptoms.

Type of participants: We included children and adolescents with documented SARS-CoV-2 infection following clinical, imaging or laboratory criteria with an assessment of symptoms or sequelae four weeks after infection, including those with asymptomatic or mildly symptomatic infection.

### 2.2. Main Outcomes

We included studies assessing the following outcomes:Health-related Quality of Life: including measurements of physical-mental-social functioning (SF-36, EuroQOL or other related scales)Changes in work/occupational and study (school attendance)Survival related to Long COVID (i.e., not overall survival related to infection, but the presence of persistent or new long-term symptoms or sequelae)Recovery/duration of symptomsNeed for rehabilitation/resource use

Timing of the outcomes: Since the acute phase of COVID-19 is usually defined by the first four weeks, persistent symptoms were defined beyond this period up to 12 weeks. Symptoms that might be attributable to post-COVID-19 condition (WHO definition) as a syndrome were included as those lasting 12 weeks or more. These symptoms were further divided into short-term (12 weeks or more up to 6 months), medium-term (more than 6 months up to 12 months) and long-term: more than 12 months.

### 2.3. Search Methods for Identification of Studies

Two independent researchers (LG and JVAF) identified the studies from our evidence map. To produce this evidence map we performed a comprehensive, systematic search with no restrictions on the language of publication or publication status on November 5th, 2021. Databases searched for the evidence map were the Cochrane-COVID-19 Study Register (https://covid-19.cochrane.org, accessed on 25 September 2022; comprising PubMed, Embase, Cochrane CENTRAL, ClinicalTrials.gov (accessed on 25 September 2022), WHO International Clinical Trials Registry Platform and medRxiv) and the WHO COVID-19 Global literature on coronavirus disease database (https://search.bvsalud.org/global-literature-on-novel-coronavirus-2019-ncov, accessed on 25 September 2022). The full methods for the search design and selection process of the evidence map are detailed elsewhere [18]. We selected the studies that met our eligibility criteria from our evidence map. Nevertheless, two study authors (GO and LG) re-assessed the eligibility criteria and we documented this selection process using a PRISMA flow diagram [21]. We also identified ongoing studies and checked for emerging results by searching for the study identification number or lead researcher’s name in Google and Pubmed.

### 2.4. Data Extraction

We developed a dedicated data abstraction form that we pilot tested using Google Spreadsheets. Two independent reviewers (LG and GO) extracted outcome data relevant to this review as needed for calculating summary statistics and measures of variance. For prevalence estimates (dichotomous data), we extracted natural frequencies or percentages with confidence intervals when available. We attempted to obtain the means and standard deviations or data necessary to calculate this information for continuous outcomes. We resolved any disagreements by discussion or, if required, by consultation with a third review author (JVAF). We contacted the authors of included studies to obtain key missing data as needed. We did not perform imputations. For studies that fulfil the inclusion criteria, two review authors (LG and GO) independently extracted the following information: bibliographic details, study design, country, setting, age (young children, school-aged children, adolescents), vaccination status, diagnosis and severity of COVID-19, gender, socioeconomic status, prognostic factors and outcomes relevant to this review.

### 2.5. Assessment of Risk of Bias in Included Studies

We assessed the risk of bias in each study using the JBI tool for prevalence studies [19]. This tool assesses nine domains, including sampling frame, sampling method, sample size, description of participants, data analysis coverage, validity and reliability of the measurement of the outcome, statistical methods and response rate.

### 2.6. Data Synthesis

Differences in the measurements of the outcomes as well as in the clinical features of study populations included in this body of research precluded the conduct of a meta-analysis. Therefore, we described the results narratively (for three of the outcomes) and produced summary tables (for one of the outcomes) with the prevalence estimates for each outcome, including proportions and mean scores [22]. We describe heterogeneity qualitatively in the discussion of our findings. Since we did not conduct meta-analysis, we were unable to conduct subgroup analysis, sensitivity analysis or statistical assessment of publication bias, but we describe the regression analysis done by individual studies considering age, gender, comorbidities, disease severity, setting and measurement of the outcome, when available. We also describe the findings from controlled studies separately, as they would allow us to define more accurate estimates of how the outcomes are related to the infection rather than contextual factors including isolation and lockdowns. We were unable to formally assess reporting biases, including publication bias using funnel plots, because no meta-analysis was conducted.

## 3. Results

We screened 9768 results from de-duplicated retrieved records to populate our evidence map. After excluding those that focused on adults and those not meeting our eligibility criteria for this review, we identified 25 studies, including over 68 thousand children and adolescents (this is an estimate since case-mix studies did not provide the exact number; see Table 1 for a summary of the main characteristics and Figure 1 for a PRISMA flow chart and see our OSF project page for the full details of the excluded studies) [23,24,25,26,27,28,29,30,31,32,33,34,35,36,37,38,39,40,41,42,43,44,45,46,47,48,49]. Most studies were available as journal articles (all in English), 84% had a longitudinal design, and six had a control group [7,23,29,30,39,41,45]. The vast majority of studies were conducted in high-income countries and included children and adolescents with mild to moderate disease from the first waves of the pandemic. None of the studies focused on socially vulnerable participants, participants with chronic conditions or those vaccinated for COVID-19. Over half of the studies (52%) included a case mix of children and adults. Eight studies described the presence of comorbidities in children and these primarily included neuropsychiatric disorders (anxiety, depression, ADHD, autism), allergic rhinitis, asthma and eczema [25,28,29,30,34,35,36,42,43]. We reported the disaggregated data for children when available. None of the included studies assessed the pre-defined outcome ‘survival’. We have also identified six additional ongoing studies on children and adolescents from our evidence map.

All studies had an adequate sample frame to address the target population, but most used convenience sampling for recruiting participants. Only five studies reported a sample size calculation analysis or had a number of participants large enough to produce a reliable estimate. Eight of the included studies provided a detailed description of the participants’ characteristics (age, sex, severity and comorbidities). For those studies with an incomplete description of participants, severity and comorbidities were the most frequent missing details. Most studies analysed the data with sufficient coverage of all identified subgroups. In almost half of the studies, some issues arose related to the methods used to identify the outcomes of interest, although all included studies assessed all participants following the same criteria. None of the studies presented serious concerns related to the statistical analyses as it was mostly simple descriptive statistics (this does not include the quality appraisal of the analysis of effect modifiers). Problems related to loss of follow-up or low response rates were identified in only a minority of studies (See Figure 2 for a summary of the quality assessment).

### 3.1. Main Outcomes

#### 3.1.1. Quality of Life

Four studies (two controlled) assessed this outcome [29,36,38,44]. The first controlled study reported data on 38,152 children and 6,630 adolescents (LongCOVIDKidsDK), most of which tested positive in the context of mild or asymptomatic disease between 4 to 9 months before this assessment, using the PedQL scale (0–100, a higher score indicates better quality of life) [29]. A second controlled study reported data on 3065 adolescents using the EQ-5D-Y, indicating the proportion with some/lots of problems related to the five dimensions of this scale [44]. The results of these two studies are summarised in Table 2 and analysed further in the section on controlled studies. The third study reported data for 38 children at six-month follow-up after Paediatric Inflammatory Multisystem Syndrome (PIMS) and indicated that 24 (65%) had no impairment, 10 (27%) had mild impairment, and 3 (8%) had severe impairment in their quality of life according to categorisations of the PedsQL scores [36]. Similar assessments were done by their parents (30 (79%), 1 (3%) and 7 (18%), respectively). Finally, a fourth study included 431 participants <60 years old of mostly non-severe cases, including only 8 adolescents (aged 10–19); however, the data for this subgroup was not available [38].

#### 3.1.2. Changes in Work/Occupation and Study

Five studies reported this outcome [26,29,34,36,40]. The LongCOVIDKidsDK controlled study indicated that 695 adolescents (10.5%) of those who tested positive for COVID-19 reported 16 or more days of school absence and poorer quality of life associated with school functioning (see Table 2 and section below on controlled studies) [29]. The subscale of the PedsQL for “school” indicated no impairment in 32 (87%), mild impairment in 2 (5%) and severe impairment in 3 (8%) children of the 38 assessed in the aforementioned study at a 6-month follow-up after Paediatric Inflammatory Multisystem Syndrome (PIMS) [36]. Another study on 518 children with non-severe illness collected parents’ perceptions of attendance to school/nursery and indicated that 36 (7%) may be spending less time in school; however, it was mostly perceived by parents as a consequence of the pandemic (79.3%) rather than illness itself (3.7%) [34].

One study with 430 mostly non-severe cases, with an estimate of fewer than 58 adolescents (based on the age distribution), highlighted that 57% of all participants presented some restriction in daily activities at a mean follow-up of 176 ± 35.1 days [26]. Another study with 990 participants included 14.6% of participants aged 1–29. This study reported that a subset of 331 participants at 10–12-month follow-up, of which 214 (68.8%) were able to resume their daily routine at a month, 77 (24.8%) at 1 to 3 months and 20 (6.4%) at 4 months or more after discharge [40].

#### 3.1.3. Recovery/Duration of Symptoms

Twenty-two studies assessed this outcome. As we found substantial heterogeneity in study design, inclusion criteria, follow-up time and definition of recovery, we summarised the data on the overall duration of symptoms and the lack of recovery following COVID-19 (as an overarching definition of people who did not recover their functional status or had persistent symptoms) in Table 3 (data from controlled studies is presented also in Table 2). Two of these studies are not included in the table because they reported this outcome for adults and children combined (i.e., not disaggregated). A small study with 17 participants with non-severe infection reported that 78% presented persistent mild cognitive deficits at a median follow-up of 78 days [47]. Another study with 116 participants with mostly non-severe infection reported that smell and taste dysfunction persisted in 32% of those affected, with complete recovery at 6 months [48].

#### 3.1.4. Need for Rehabilitation/Resource Use

Four studies reported some data on this outcome. A large controlled study using insurance data with 57,763 children and adolescents reported an incidence rate of 436 new diagnoses/1000 persons-year, following COVID-19 compared to 335 among controls [41] (see Table 2 and the section below). The study on 46 children with PIMS indicated that four patients were readmitted: one for new-onset encephalopathy and three for infectious complications [36]. Another study on 50 children with PIMS highlighted that four children required specialist assessment and interventions due to persistent dysphagia [27]. Finally, one study included 3677 participants, of whom a minority (<25%) were adolescents and young adults and reported readmission rates; however, the age range for those readmitted excluded adolescents [31].

#### 3.1.5. Controlled Studies (Comparison with Seronegative Participants and Other Infections)

A summary of the main findings from the three largest controlled studies is in Table 2. The LongCOVIDKidsDK population-based nationwide cohort study compared data from 38,152 children and 6630 adolescents who tested positive for COVID-19 versus a control group of 147,212 children and 21,640 adolescents with negative or no testing. This study indicated a greater prevalence of symptoms (61.9% vs. 57%, OR 1.22 95% CI 1.15 to 1.30) and a greater proportion of ≥16 days of school absenteeism in those who tested positive compared to the control group [29,30]. Nevertheless, this study found higher quality of life measurements across all domains, although these differences were not clinically meaningful (Hedges < 0.2) [29,30]. Finally, a large cohort study using insurance data from 57,763 children and adolescents with COVID-19 diagnosis matched with controls found a higher incidence of new health problems recorded in health records, as a proxy for healthcare utilisation (incidence rate ratio 1.30, 95% CI 1.25 to 1.35) [41].

One smaller school-based retrospective study included 109 COVID-19 seropositive and 1246 seronegative children and adolescents and highlighted a similar proportion of symptoms lasting >4 and >12 weeks (*p* value not available) [39].

Two controlled studies presented relevant data but were not disaggregated for the subgroup of children and adolescents. One study reported the persistence of symptoms in a cohort of 293 participants (16 children) and highlighted that those with confirmed infection had a higher persistence of symptoms compared to seronegative exposed controls (*p*-value < 0.05) [23]. The other study based on medical records included data from 273,618 COVID-19 survivors, of which 29,753 were aged 10–21, matched with a control of influenza survivors. This study reported that the incidence of each and any Long COVID feature was higher after COVID-19 than symptom persistence following influenza (HR 1.65, 95% CI 1.62 to 1.67) [45].

### 3.2. Effect Modifiers

The LongCOVIDKidsDK study identified that more female participants had persistent symptoms in the case and in the control group for those 12–18 years, but not for younger age groups (12–14 years OR 1.70, 95% CI 1.47 to 1.97 cases, OR 1.47 95% CI 1.36 to 1.59 controls; 15–18 years: OR 2.70 95% CI, 2.40 to 3.03 cases and OR 2.56 95% CI 2.42 to 2.70 controls) [29,30]. The match-controlled CLoCk study found that those with a higher burden of symptoms had poorer previous physical and mental health, were older (in the category 15–17 years) and were more often female; however, this was also true for both the test-negative and the test-positive groups [44]. This study also collected data on deprivation (as a proxy for socioeconomic status), but no analysis was presented for this variable. The large controlled study on insurance data identified that the incidence of new diagnosis as a proxy for resource utilisation was higher in those with hospitalisation and intensive care compared to outpatients; however, these inferences were underpowered for the subgroup of children and adolescents [41].

One uncontrolled study focusing exclusively on children and including 518 participants performed multivariable logistic regression to investigate the association of demographic characteristics with persistent symptoms [34]. The study showed that older age was associated with persistent symptoms, especially when comparing children older than 6 years with those under 2 years old: For every child under 2 years of age with persistent symptoms, there were almost three children older than 6 years with persistent symptoms (OR 2.68, 95% CI 1.41 to 5.4). Allergic comorbidities were also positively associated with persistent symptoms (OR 2.66, 95% CI 1.04 to 6.47) and neurological conditions (OR 4.38, 95% CI 1.36 to 15.67) [49]. No association between gender and persistent symptoms was identified [34]. We found no additional information related to other predefined subgroups, including socially vulnerable individuals or people with other comorbidities.

### 3.3. Ongoing Studies

We identified six ongoing studies. One cohort study aims to describe the follow-up of children presenting at emergency departments at 14 and 90 days [50,51]. Five records are studies registers addressing different long-term functional characteristics or impacts of SARS-CoV 2 infection in children, two of which specified quality of life measurements (see https://osf.io/b7dwy/, accessed on 25 September 2022).

## 4. Discussion

### 4.1. Main Findings

Most of the studies were conducted in high-income countries, and over half (52%) included a case mix of unvaccinated children and adults in the earlier phases of the pandemic. Data on recovery rates are highly heterogeneous, ranging from 54% to 95% at six-month follow-ups and 49% to 92% at 12 months. Most children with persistent symptoms reported mild or no impairment in their quality of life at a 6-month follow-up. Most children with persistent symptoms reported no substantial impairment in their school functioning at 3–6 month follow-ups, although their parents indicated their opinion that their children may be spending less time in school as a consequence of the pandemic rather than the illness itself. Hospital readmission and interventions due to persistent dysphagia following PIMS were the most frequently reported rehabilitation interventions related to resource use. The certainty of the evidence for these outcomes is very limited. Most studies presented some concerns related to study design, including the recruitment of participants, the sample size (mostly small) and the description of study subjects and setting. Heterogeneity in study populations and how outcomes were measured and reported prevented us from conducting a meta-analysis.

### 4.2. Related Research

Two previous systematic reviews on Long COVID in children and adolescents reported a wide range of symptoms following acute SARS-CoV-2 infection, including respiratory, neurological or cognitive symptoms [16,17]. In the review by Behnood and colleagues, primary meta-analyses were conducted on the prevalence of persistent individual symptoms, based on 8 studies including control groups, and secondary analyses included a total of 22 studies identified up to 31 July 2021 [17]. The review by Lopez-Leon and colleagues included a meta-analysis of symptom prevalence based on all 21 studies identified up to 10 February 2022. These authors did not stratify analyses by study design and pointed out the considerable risk of bias in particular because of the lack of standardised definitions of symptoms and a high level of heterogeneity [16]. Two recent systematic reviews including studies up to early 2022 reported rates of post-COVID syndrome in children ranging from 0% to 70% [52,53]. One of these reviews highlighted the critical risk of bias across studies, mostly due to confounding [52]. Regarding healthcare resources use, Magnusson et al. conducted a before and after study to explore if the use of healthcare services is mildly increased among children and adolescents after COVID-19, mostly in primary care settings due to respiratory and general unspecified conditions during the first months, with limited impact on healthcare services, especially in children under 5 years of age [54]. Although the scope of our review relates to children and adolescents, similar limitations in the body of research were found for adults. For instance, a recent umbrella review of 18 systematic reviews indicated that few studies reported the quality of life in adults, yielding heterogeneous results across mostly uncontrolled studies with a high risk of bias [55].

Several large ongoing studies aim to provide insight into the long-term health impacts on children and adolescents following SARS-CoV-2 infection. Implementing follow-up questionnaires, the COVID-19 Schools Infection Survey, England provides regular estimates of the proportions of children and adolescents 3–16 years who have symptoms persisting for at least 12 months after testing positive for SARS-CoV-2 infections well as proportions who are impacted by these symptoms in everyday life [56]. The British CLoCk study will continue to follow children and adolescents 11–17 years of age at months 6 and 9 following SARS-CoV-2 infection compared to matched controls [44].

Our findings highlight the need for continued research in order to understand and quantify the health impact of COVID-19 among children and adolescents and to assure appropriate health care and social services. Including control groups is essential in order to identify health impacts directly related to Sar-CoV-2 infection rather than other contextual factors. However, given the increasing numbers of children and adolescents getting infected and a higher chance of asymptomatic or paucisymptomatic COVID-19 among children, in particular small children than among adults, it will be increasingly challenging to choose appropriate control groups. Above all, the choice and definition of health endpoints will need harmonisation, even more so with regard to research on Post-COVID-19 conditions among children and adolescents than among adults [57].

### 4.3. Limitations

We empirically derived a search strategy with high sensitivity to retrieve reports on post-COVID-19 conditions, but it might be less sensitive for reports focused on more specific sets of complications. Moreover, smaller studies and poorly described ones might not have been picked up by the search. However, many Cochrane reviews used this set of comprehensive resources [58,59,60], and it is unlikely that we missed relevant larger and controlled studies. Additionally, data extraction resulted in challenges due to the poor reporting of included studies. In some cases, we had to infer study design (longitudinal, as those studies with at least two timepoints for assessment), the severity of the included study population and reported outcomes. Considering the lack of study registration of most included studies, this poses challenges in assessing the validity of reported results. Finally, we were unable to pool data due to the heterogeneity of the study population and outcome definitions, which precluded the estimation of the total number of affected participants across studies for each outcome.

## 5. Conclusions

There is limited evidence on the short and long-term well-being of children following SARS-CoV-2 infection. The findings from these first studies indicate that long-term symptoms may be present in children, but health impact estimates were heterogeneous. High-quality longitudinal studies with defined health outcomes and valid control groups are needed to understand and quantify the direct and indirect health impacts of the COVID-19 pandemic on children and adolescents and to provide appropriate health care and counselling services. This is particularly important as original research studies need to cover the later stages of the pandemic following the implementation of vaccination and the emergence of new variants as well as reinfections.

## Figures and Tables

**Figure 1 ijerph-19-14392-f001:**
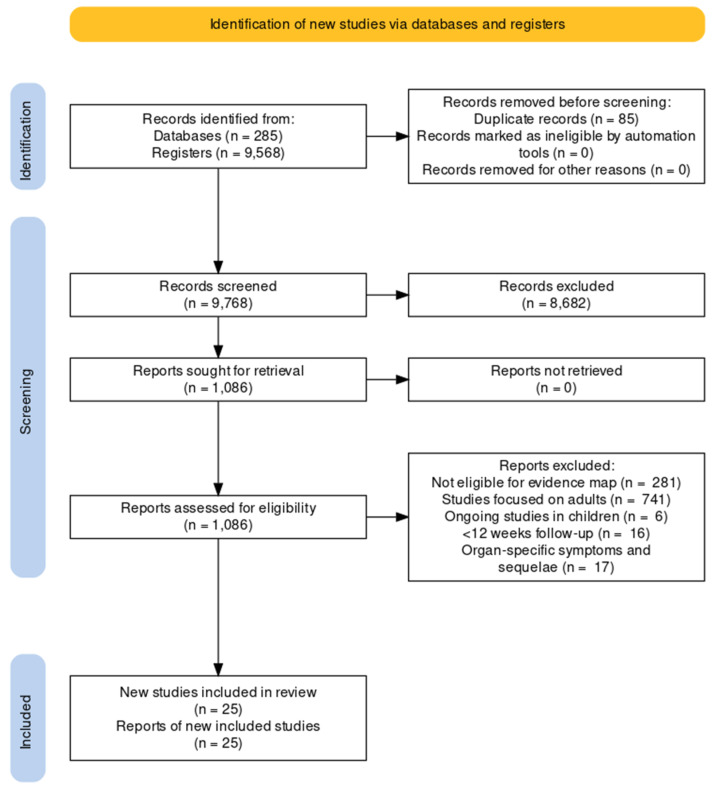
PRISMA flow diagram.

**Figure 2 ijerph-19-14392-f002:**
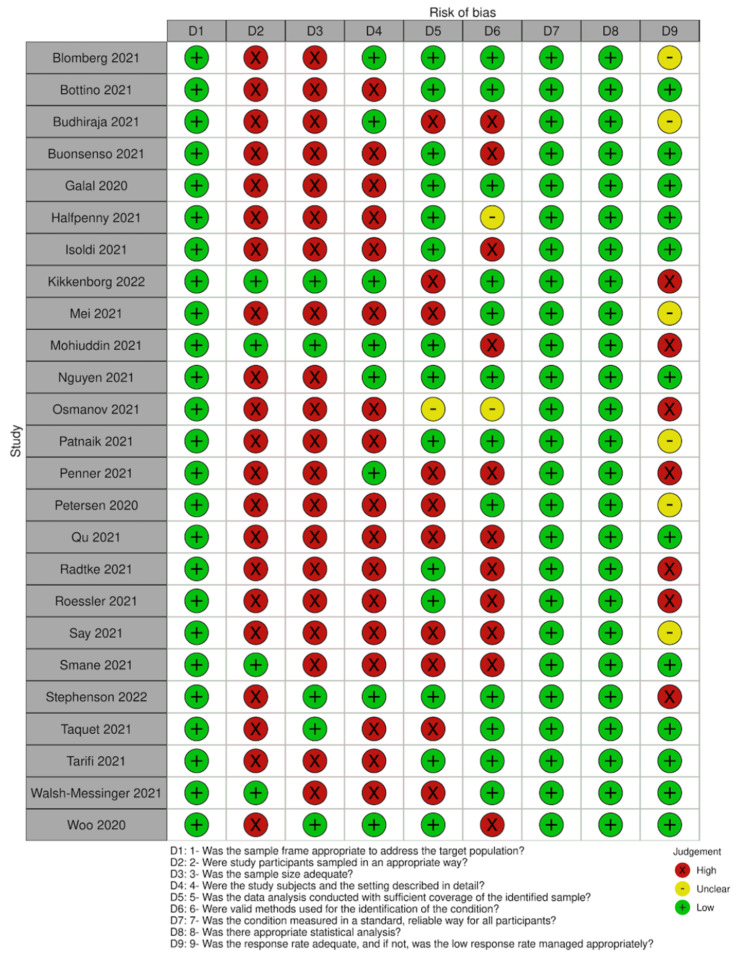
Summary of quality assessment.

**Table 1 ijerph-19-14392-t001:** Characteristics of included studies.

Characteristics	Proportion
Study Design	
Cross-sectional	4/25 (16%)
Longitudinal	21/25 (84%)
With a control group	6/25 (24%)
Median sample size (interquartile range)	200 participants (92 to 990)
Setting	
Country	
High income	17/25 (68%)
Upper middle income	4/25 (16%)
Lower middle income	4/25 (16%)
Recruitment	
Community/contact tracing	7/25 (28%)
Outpatient	9/25 (36%)
Hospital	13/25 (52%)
ICU	5/25 (20%)
Population	
Children	
Aged 0–5	14/25 (56%)
Aged 6–11	19/25 (76%)
Aged 11–18	22/25 (88%)
Severity	
Asymptomatic	8/25 (34.78%)
Mild	17/25 (73.91%)
Moderate	12/25 (52.17%)
Severe	11/25 (47.83%)
Critical	10/25 (43.48%)

**Table 2 ijerph-19-14392-t002:** Summary of the findings from the main large controlled studies.

Outcome/Study	Cases Mean ± SD/% (n)	Controls Mean ± SD/% (n)
Quality of Life		
LongCOVIDKidsDK (cross-sectional >4 months)—physical functioning—PedsQL [29,30]		
1–12 months	93.7 ± 11.2 (105)	87.8 ± 12.2 * (348)
13–24 months	94.2 ± 9.1 (427)	87.3 ± 12.0 * (1062)
2–3 years	94.8 ± 10.2 (917)	94.8 ± 8.2 (2445)
4–11 years	94.7 ± 11.4 (6032)	92.9 ± 11.8 * (18,372)
12–14 years	93.0 ± 13.0 (3516)	91.2 ± 13.3) * (10,789)
15–18 years	88.7 ± 13.9 (6630)	86.5 ± 14.3 (21,640)
LongCOVIDKidsDK (cross-sectional >4 months)—emotional functioning—PedsQL [29,30]		
1–12 months	75.5 ± 16.9 (105)	75.8 ± 13.7 (348)
13–24 months	73.6 ± 16.2 **** (427)	77.0 ± 12.8 **** (1062)
2–3 years	75.5 ± 18.1 (917)	73.5 ± 15.4 (2445)
4–11 years	78.2 ± 19.1 **** (6032)	73.3 ± 18.0 **** (18,372)
12–14 years	83.2 ± 19.5 **** (3516)	79.2 ± 19.2 **** (10,789)
15–18 years	77.1 ± 20.3 (6630)	71.7 ± 21.4 (21,640)
LongCOVIDKidsDK (cross-sectional >4 months)—social functioning—PedsQL [29,30]		
1–12 months	94.7 ± 9.3 (105)	93.0 ± 11.4 (348)
13–24 months	93.3 ± 11.0 (427)	93.0 ± 9.9 (1062)
2–3 years	93.8 ± 10.8 * (917)	93.0 ± 10.8 * (2445)
4–11 years	92.3 ± 13.3 * (6032)	89.6 ± 15.0 * (18,372)
12–14 years	91.4 ± 15.4 **** (3516)	87.9 ± 17.5 **** (10,789)
15–18 years	93.1 ± 12.5 (6630)	88.4 ± 16.2 (21,640)
CLoCk study—matched cohort study—3 months—11 to 17 years—EQ-5D-Y ** [44]		
Problems with mobility	4.0/16.9% (3017/722)	4.1/11.9% (2158/907)
Problems with self-care	2.4/9.0% (3017/722)	3.6/10.3% (2158/907)
Problems with usual activities	9.4/34.4% (3017/722)	9.5/31.1% (2158/907)
Pain/discomfort	8.6/40.1% (3017/722)	8.9/41.8% (2158/907)
Worried/sad/unhappy	31.4/64.9% (3017/722)	32.9/69.4% (2158/907)
Changes in School		
LongCOVIDKidsDK (cross-sectional >4 months)—school functioning—PedsQL [29,30]		
2–3 years	92.9 ± 12.1 (917)	93.0 ± 11.2 (2445)
4–11 years	86.8 ± 15.3 (6032)	84.2 ± 15.4 ** (18,372)
12–14 years	83.7 ± 18.0 (3516)	80.9 ± 17.8 ** (10,789)
15–18 years	66.9 ± 22.6 (6630)	62.9 ± 22.1 *** (21,640)
LongCOVIDKidsDK—16 days or more of school absence [29,30]		
13 months—3 years	23.9% (1062)	14.1% ** (3507)
4–11 years	6.1% (6032)	3.3% ** (18,372)
12–14 years	6.5% (3516)	5.0% ** (10,789)
15–18 years	10.5% (6630)	8.2% ** (21,640)
Recovery		
LongCOVIDKidsDK—cross-sectional—overall persistence of symptoms >2 months [29,30]		
0–3 years	40.0% (1194)	27.2% (3855) ***
4–11 years	38.1% (5023)	33.7% (18,372) ***
12–14 years	46.0% (2857)	41.3% (10,789) ***
15–18 years	61.9% (6630)	57.0% (21,640) ***
CLoCk Study—cohort study—persistence of symptoms at 3 months [44]		
11–14 years	60.5% (1244)	47.5 (1609) **
15–17 years	70.6% (1821)	57.7% (2130) **
Probability of multiple symptoms	29.6% (3065)	19.3% (3739) **
Resource Utilisation		
(Insurance data) Incidence rate of new diagnosis >3 months [41]	436.91/1000 persons-year	335.98/1000 persons-year **

* *p* value < 0.05 but small effect size (Hedges < 0.2)—** This study reported proportions in two subgroups: those with fewer symptoms/those with symptoms. *** *p* < 0.05 compared to cases. **** *p* value < 0.05 but considerable effect size (Hedges > 0.2).

**Table 3 ijerph-19-14392-t003:** Summary of studies reporting recovery/duration of symptoms.

	Clinical Features		n Children Analysed	% Unrecovered (with Persistent Symptoms) or Duration			
	Severity	Comorbidities		4–8 Weeks	8–12 Weeks	3–6 Months	6–12 Months
Cohort Studies: Overall Recovery from Symptoms							
Blomberg 2021 [23]	22% hospitalised, 3% severe	44% *	16			13%	
Bottino 2021 [24]	Non-severe cases	N/A	16	median 67 days (range 49–91)			
Isoldi 2021 [28]	Non-severe cases	27%	15	0%			
Mohiuddin 2021 [32]	20% hospitalised	20% *	22		23%	5%	
Osmanov 2021 [34]	37% pneumonia, 3% severe	27%	518			20%	11%
Petersen 2020 [37]	4% hospitalised *	N/A **	21			30%	
Radtke 2021 [39]	Non-severe cases	N/A	109			4%	
Budhiraja 2021 [40]	23% moderate 15% severe **	37% *	145		38%		8%
Say 2021 [42]	Non-severe cases except 2 PIMS	N/A **	171			8%	
Smane 2021 [43]	Non-severe cases	20%	92	55%			
Stephenson 2022 [44]	Not specified	N/A	3065			61–71%	
Taquet 2021 [45]	Not specified	N/A **	29,753			46%	
Cohort Studies: Recovery in Subpopulations/Specific Symptoms							
Halfpenny 2021 [27]	PIMS—dysphagia	N/A **	50	median 45.5 days (range 28–127)			
Patnaik 2021 [35]	PIMS	N/A **	21		0%		
Penner 2021 [36]	PIMS	17%	46			45%	
Cross-Sectional: Overall Recovery from Symptoms							
Buonsenso 2021 [25]	Mostly non-severe, 4.7% hospitalised, 2.3% critical	N/A **	129	65%	67%		51%
Kikkenborg 2022 [29,30]	14–18 years: Mostly mild or asymptomatic, 9% severe	NA **	6630		62%		
	0–14 years: Mostly mild or asymptomatic, <3.5% severe	NA **	38,152		40–46%		
Walsh-Messinger 2021 [46]	Mostly non-severe, 5% severe	N/A	<26	30%			
Galal 2020 [26]	Hospitalisation 24%, oxygen therapy 17%, ICU 5%		<58	mean 176 ± 35.1 days			
Cross-Sectional: Recovery in Subpopulations/Specific Symptoms							
Nguyen 2021 [33]	Non-severe cases—anosmia/dysgeusia	N/A **	<50	~30%			

Notes: * Includes population aged 10–21—** only disaggregated data for each comorbidity—*** Estimated based on the distribution of age—PIMS: Paediatric Inflammatory Multisystem Syndrome—ICU: Intensive Care Unit.

## Data Availability

Additional description of the initial protocol and subsequent amendments, the included, excluded and ongoing studies can be found at the OSF project site (https://osf.io/b7dwy/, accessed on 25 September 2022) and in the evidence map available at the RKI website (www.rki.de/post-covid-evimaps, accessed on 25 September 2022).

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
