# Peer review of "Short and Long-Term Wellbeing of Children following SARS-CoV-2 Infection: A Systematic Review"

_ijerph, 2022, doi:10.3390/ijerph192114392_

Round 1

Reviewer 1 Report

Abstract

·         Please provide further information about the study outcomes

·         In the abstract you need to answer the following questions, what, why and how and discuss the study new findings, limitations, and future research

·         The abstract should state briefly the purpose of the research, the principal results and major conclusions. An abstract is often presented separately from the article, so it must be able to stand alone

Introduction

-          discuss the research aims, research gap and discuss the paper layout Add up-to-date references to support your discussion

-          The necessity and innovation of the article should be presented to the introduction.

Methods

·         How the authors treat the heterogeneous of the studies methodologies? Did the authors used random effect approaches ? 

·    

·   

Discussion

-          I believe that more in depth discussion is needed. The discussion as present now is simple and concise. Revision of more papers using similar technique is needed

-          In the discussion, please discuss if the study research questions are answered or not Also introduce the model in detail. Draw a conclusion from this study and present the limitations and future research.

-          . The major defect of this study is the debate or Argument is not clear stated in the introduction session. Hence, the contribution is weak in this manuscript. I would suggest the author to enhance your theoretical discussion and arrives your debate or argument

-          Please make sure your conclusions' section underscore the scientific value added of your paper, and/or the applicability of your findings/results, as indicated previously.

-          Please revise your conclusion part into more details. Basically, you should enhance your contributions, limitations, underscore the scientific value added of your paper, and/or the applicability of your findings/results and future study in this session.

Author Response

Abstract

Please provide further information about the study outcomes

Response: we currently describe the outcomes in the abstract (see in bold)

We included observational studies on children and adolescents with a follow-up greater than 12 weeks and focused on the outcomes quality of life, recovery/duration of symptoms, school attendance and resource use/rehabilitation. We assessed their methodological quality, and we prepared a narrative syn-thesis of the results. We included 21 longitudinal and 4 cross-sectional studies (six with a control group) with over 68 thousand unvaccinated children and adolescents with mostly asymptomatic or mild disease.  Study limitations included convenience sampling, poor description of their study population and heterogeneous definitions of outcomes. Quality of life was not largely affected in adolescents following COVID-19, but there might be greater impairment in young children and in those with more severe forms of the disease (4 studies). There might also be an impairment in daily activities and increased school absenteeism following COVID-19, but the findings were hetero-geneous (5 studies). 22 studies provided highly variable estimates based on heterogeneous defi-nitions of overall persistence of symptoms (recovery), ranging from 0 to 67% at 8-12 weeks and 8 to 51% at 6-12 months. We found limited data on resource use and the need for rehabilitation. One controlled study indicated that the quality of life of infected children and adolescents might not substantially differ from controls. All controlled studies found a higher burden of persistent symptoms in COVID-19 cases compared with test-negative controls or cases of influenza. (Page 1 - Abstract)

  •         In the abstract you need to answer the following questions, what, why and how and discuss the study new findings, limitations, and future research
  •         The abstract should state briefly the purpose of the research, the principal results and major conclusions. An abstract is often presented separately from the article, so it must be able to stand alone

Response to the two points above: We currently describe:

  1. The purpose of the research: We aimed to describe the findings of studies assessing key outcomes related to global wellbeing and recovery in children and adolescents from a public health perspective.
  2. The principal results (see above)
  3. Conclusions: There is limited evidence on the short and long-term well-being of children following SARS-CoV-2 infection. 
  4. Recommendations for further research: High-quality longitudinal studies with control groups are needed to describe the outcomes in this population, especially in vaccinated children and those affected by new variants of the virus.

Introduction

Discuss the research aims, research gap and discuss the paper layout Add up-to-date references to support your discussion. The necessity and innovation of the article should be presented to the introduction.

Response: thank you, we rephrased this here at the end of the introduction:

Our team has developed an evidence map of observational studies analysing long-term symptoms and sequelae following SARS-CoV-2 infection, which is available on the website of the Robert Koch Institute (www.rki.de/post-covid-evimaps). Up to November 2021, we found few studies that assessed the long-term course and prognosis of infection clinically in children and adolescents, as they mostly focused on the burden of individual symptoms (e.g., anosmia, fatigue, etc.). In this systematic review, we aim to describe the findings of studies assessing key outcomes related to global wellbeing and recovery in children and adolescents using the evidence derived from our evidence map[18]. (page 2 - lines 75-83)

Methods

  •         How the authors treat the heterogeneous of the studies methodologies? Did the authors used random effect approaches ?

Response: We did not conduct metaanalysis: “Differences in the measurements of the outcomes as well as in the clinical features of study populations included in this body of research precluded the conduct of a me-ta-analysis.”

Discussion

-          I believe that more in depth discussion is needed. The discussion as present now is simple and concise. Revision of more papers using similar technique is needed

Response: We expanded our analysis of more recent papers: “Two previous systematic reviews on Long COVID in children and adolescents reported a wide range of symptoms following acute SARS-CoV-2 infection, including respiratory, neurological or cognitive symptoms[17,52]. In the review by Behnood and colleagues, primary meta-analyses were conducted on the prevalence of persistent individual symp-toms, based on 8 studies including control groups, and secondary analyses included a to-tal of 22 studies identified up to 31 July 2021 [17]. The review by Lopez-Leon and col-leagues included a meta-analysis on symptom prevalence based on all 21 studies identi-fied up to February 10, 2022. These authors did not stratify analyses by study design and pointed out the considerable risk of bias in particular because of the lack of standardised definitions of symptoms and a high level of heterogeneity [52]. Two recent systematic re-views including studies up to early 2022 reported rates of post-COVID syndrome in chil-dren ranging from 0% to 70% [53-54]. One of these reviews highlighted critical risk of bias across studies mostly due to confounding [53]. Regarding healthcare resources use, Mag-nusson et al. conducted a before and after study to explore if the use of healthcare services is mildly increased among children and adolescents after COVID-19, mostly in primary care settings due to respiratory and general unspecified conditions during the first months, with limited impact on healthcare services, especially in children under 5 years of age[55]. Whereas the scope of our review relates to children and adolescents, similar limitations in the body of research were found for adults. For instance, a recent umbrella review of 18 systematic reviews indicated that few studies reported the quality of life in adults, yielding heterogeneous results across mostly uncontrolled studies with a high risk of bias [56].” (page 14 - line 347-368)

-          In the discussion, please discuss if the study research questions are answered or not. Also introduce the model in detail. Draw a conclusion from this study and present the limitations and future research.

-          Please make sure your conclusions' section underscore the scientific value added of your paper, and/or the applicability of your findings/results, as indicated previously.

-          Please revise your conclusion part into more details. Basically, you should enhance your contributions, limitations, underscore the scientific value added of your paper, and/or the applicability of your findings/results and future study in this session.

Response to the three comments above: Thanks for highlighting this. We have not presented a model in our research. Our current conclusions include the findings of the study and future research “There is limited evidence on the short and long-term well-being of children following SARS-CoV-2 infection. The findings from these first studies indicate that long-term symp-toms may be present in children, but health impact estimates were heterogeneous. High-quality longitudinal studies with defined health outcomes and valid control groups are needed to understand and quantify the direct and indirect health impacts of the COVID-19 pandemic on children and adolescents and to provide appropriate health care and counselling services. This is particularly important as original research studies need to cover later stages of the pandemic following the implementation of vaccination and the emergence of new variants as well as reinfections.” (page 16 lines 402-410)

-          . The major defect of this study is the debate or Argument is not clear stated in the introduction session. Hence, the contribution is weak in this manuscript. I would suggest the author to enhance your theoretical discussion and arrives your debate or argument

Response: We currently present the state-of-the-art definitions of Long-COVID in our introduction and the need for further development. This is an evolving field of research.

Reviewer 2 Report

Overall, I think this is an excellent study. There are only minor comments. 

1) The PRISMA flow diagram is incomplete. Please refer to PRISMA and revise the diagram. 

2) Remove the extra ")" in Table 2

Author Response

Overall, I think this is an excellent study. There are only minor comments. 

1) The PRISMA flow diagram is incomplete. Please refer to PRISMA and revise the diagram. 

Response: We replaced the PRISMA flow diagram with a full explanation including the screening from the evidence map.

2) Remove the extra ")" in Table 2

Response: Thanks for spotting this. We removed the extra “)”

Reviewer 3 Report

The present review is detailed and carried out quite seriously and methodically.

A few additional points:

1.  Is there any specific or estimated information about the VACCINATION status (percentage?) of the children/ /adolescents?

2.         The total impression of the report would have been much more clear and conceivable if specific or even roughly estimated numbers about the effect of COVID-19 on the children/adolescents were TOTALLY reported.

Author Response

The present review is detailed and carried out quite seriously and methodically.

A few additional points:

  1. Is there any specific or estimated information about the VACCINATION status (percentage?) of the children/ /adolescents?

Response: We found no studies on vaccinated children and adolescents - “None of the studies focused on socially vulnerable participants, participants with chronic conditions or those vaccinated for COVID-19.” (page 4, lines 174-176)

  1.         The total impression of the report would have been much more clear and conceivable if specific or even roughly estimated numbers about the effect of COVID-19 on the children/adolescents were TOTALLY reported.

Response: You are right, however we are limited by the characteristics of the evidence base. We added: “Finally, we were unable to pool data due the heterogeneity of study population and outcome definitions, which precluded the estimation of the total number of affected participants across studies for each outcome.” (Page 15, lines 398-400)